# Diversity, Biosynthesis and Bioactivity of Aeruginosins, a Family of Cyanobacteria-Derived Nonribosomal Linear Tetrapeptides

**DOI:** 10.3390/md21040217

**Published:** 2023-03-29

**Authors:** Jiameng Liu, Mengli Zhang, Zhenkuai Huang, Jiaqi Fang, Zhongyuan Wang, Chengxu Zhou, Xiaoting Qiu

**Affiliations:** 1Ministry of Education Key Laboratory of Applied Marine Biotechnology, Ningbo University, Ningbo 315800, China; 2Institute of Marine Biotechnology, College of Food and Pharmaceutical Sciences, Ningbo University, Ningbo 315800, China; 3Li Dak Sum Yip Yio Chin Kenneth Li Marine Biopharmaceutical Research Center, Ningbo University, Ningbo 315800, China

**Keywords:** aeruginosins, Choi, biogenesis, structural diversity, nonribosomal polypeptide synthesis, serine protease inhibitory activity

## Abstract

Aeruginosins, a family of nonribosomal linear tetrapeptides discovered from cyanobacteria and sponges, exhibit in vitro inhibitory activity on various types of serine proteases. This family is characterized by the existence of the 2-carboxy-6-hydroxy-octahydroindole (Choi) moiety occupied at the central position of the tetrapeptide. Aeruginosins have attracted much attention due to their special structures and unique bioactivities. Although many studies on aeruginosins have been published, there has not yet been a comprehensive review that summarizes the diverse research ranging from biogenesis, structural characterization and biosynthesis to bioactivity. In this review, we provide an overview of the source, chemical structure as well as spectrum of bioactivities of aeruginosins. Furthermore, possible opportunities for future research and development of aeruginosins were discussed.

## 1. Introduction

Water eutrophication is a phenomenon of water pollution caused by the enrichment of nutrients containing phosphorus and nitrogen in water, which is manifested by the abnormal reproduction and growth of algae and other plankton, the reduction of dissolved oxygen in water and the death of a large number of aquatic organisms [1]. Due to eutrophication, a large number of algae, with cyanobacteria and green algae as the dominant species, grow on the surface of the water, forming a “green scum”—water bloom. This results in the release of large amounts of harmful gases from the accumulation of organic matter in the bottom layer under anaerobic conditions as well as the release of large amounts of algal toxins due to the rupture of algal cells, posing a serious threat to the safety of drinking water for humans and animals [2].

Cyanobacterial blooms are distributed in tropical, subtropical and temperate regions of the world [3]. China is one of the countries where cyanobacterial blooms occur most severely in the world. Although water blooms cause a variety of hazards, cyanobacteria that cause water blooms contain rich and diverse secondary metabolites, so they are considered to be important sources of drug candidates and precursors. Among the different cyanobacteria-derived secondary metabolites identified, aeruginosins, a class of bioactive tetrapeptides that appear during cyanobacterial blooms in natural waters, have been found in *Microcystis* [4,5,6,7,8], *Planktothrix* [9,10], *Nostoc* [11,12], *Nodularia* [13,14] of the cyanobacterial phylum, and were found in sponges (probably in symbiotic cyanobacteria) as well (Table 1) [15,16,17]. The structure of this peptide was initially elucidated by two-dimensional nuclear magnetic resonance (2D-NMR) during the screening of metabolites from *Microcystis aeruginosa*, which marked the discovery of a new class of peptides possessing serine protease inhibitory activity [18,19]. Aeruginosin is characterized by the occupation of the 2-carboxy-6-hydroxy-octahydroindole (Choi) moiety at the central position in the tetrapeptide, while other positions are occupied by a single variable residue of congeners [20]. Aeruginosins are a family of chemo-diverse peptides that have been shown to inhibit serine protease in vitro [21]. The mechanism of inhibition has been elucidated by X-ray crystallographic analysis of the structure of the aeruginosin–protease complex [19,22]. Considering the potential druggability of aeruginosin, its organic synthesis has been carried out. Aeruginosin 298A is the first member of the aeruginosin family isolated by Murakami and his team in 1994, so it is also the original synthetic target selected by many researchers [23]. After the efforts of many years, the organic syntheses of aeruginosin 298A and 298B were finally completed in 2001 based on the absolute configuration observed in the crystal structure [18,24]. As the research progressed, total syntheses of microcin SF-608 [6], chlorodysinosin A [25], oscillarin [26] and aeruginosin KT608A [27] were also completed.

Aeruginosins have attracted much attention due to their special structures and unique bioactivities. Although these compounds have similar backbones, the isolation of new members of the aeruginosin family is often accompanied by the discovery of new residues, especially arginine derivatives [19].

## 2. Biogenesis

### 2.1. Genus Microcystis

In the early 1990s, aeruginosin 298A (Figure 1) was isolated from *Microcystis aeruginosa* (strain NIES-298). It is the first member of the aeruginosin family that has ever been identified, and its structure was originally determined by 2D-NMR [18]. In 1998, the crystal structure of the complex of aeruginosin 298A with leech thrombin was determined at 2.1 Å resolution, leading to the confirmation of an absolute stereo configuration that revealed some unexpected interactions, which could be utilized for structure-based drug design [28].

In 1995, Murakami and colleagues isolated the new trypsin inhibitors aeruginosin 98A and 98B (Figure 1) from the cultured freshwater cyanobacterium *Microcystis aeruginosa* (NIES-98) [22]. The absolute configurations of the peptides were determined by acid hydrolysis of the peptides, derivatization and subsequent chiral phase high-performance liquid chromatography (HPLC) with standard samples as references, combined with 2D-NMR analysis. The absolute configuration of the Choi moiety was determined by derivatization and NMR analysis. The absolute configuration of aeruginosin 98B was determined by X-ray crystallographic analysis of its ternary complexes with hirudin and thrombin [22].

At the end of the 20th century, aeruginosin 98C and 298B (Figure 2) were isolated from *Microcystis aeruginosa* (strains NIES-298 and NIES-98). In addition, three new congeners, aeruginosin 101, 89A and 89B (Figure 2), were isolated from other algal strains (NIES-101 and NIES-89). Their structures were determined by 2D-NMR. In HPLC, it was found that aeruginectin 89A and 89B were tautomeric, containing argininal with a stereo configuration of L-type or D-type [4]. In the same year, Carmeli and coworkers discovered microcin SF608 (Figure 2) during screening for protease inhibitors from a non-toxic strain of *Microcystis aeruginosa*. Through 2D-NMR and HPLC analysis, the structure of microcin SF608 (9) (Figure 2) was elucidated, and it was found to be very similar to that of aeruginosin 298A (Figure 1) [7].

In August 2007, Elkobi-Peer and colleagues extracted a freeze-dried water extract of *Microcystina aeruginosa* collected from a fish pond in Kibbutz Geva, Israel, with 70% methanol. Five novel natural products, aeruginosin GE686, GE766, GE730, GE810 and GE642 (14) (Figure 3) [5], as well as four known aeruginosins: 98C, 101 (Figure 2) [4], KY642 (Figure 3) [7] and DA688 (Figure 4) [8], were obtained. Various spectroscopic techniques, including NMR and mass spectrometry, were used to identify the structure, while the absolute configuration of the center was determined by Marfey’s method and chiral phase HPLC.

### 2.2. Genus Planktothrix

In 1997, Shin and colleagues isolated aeruginosins in *Planktothrix agardhii* (NIES-205) from cyanobacteria collected from Lake Kasumigaura, Japan. Extracts with significant inhibitory activities on trypsin and thrombin were obtained by mass cultivation, and two new types of aeruginosins, aeruginosin 205A and 205B (Figure 4), were purified by reverse HPLC by using Cosmosil C-18 column [9]. Since the data of their 2D-NMR spectra are basically identical, it can be assumed that they have the same stereochemistry.

In the same year, oscillarin (Figure 5), a new type of aeruginosin, was isolated from the cultures of *Planktothrix agardhii* (strain B2 83) [10]. Based on NMR data and the crystal structure of its complex with trypsin, the structure and absolute configuration were confirmed. Oscillarin is composed of D-phenyllactic acid (D-Pla), D-Phe, L-Choi and the cyclic guanidine. In 2004, Hanessian and colleagues succeeded in obtaining a complex of oscillarin and α-thrombin-huridin (Hirudin is a 65 amino acid residue protein isolated from the salivary glands of the medicinal leech *Hirudo medicinalis*. Hirugen is a close analog of the C terminus of this recombinant form of hirudin. Since hirugen prevents proteolysis of thrombin in vitro, it was possible to obtain crystals of the hirugen–thrombin complex without autolysis. Importantly, the residues of the catalytic triad in this active structure and those in the hirudin complexed structure, which is in an inactive state, are remarkably similar [32].), which can be structurally resolved by X-ray diffraction at 2.0 Å resolution [26]. The structure of the thrombin–oscillarin complex was further confirmed by total synthesis and high-resolution X-ray diffraction data, and the revised structure revealed the presence of 1-amino-2-(N-amidino-Δ^3^-pyrrolinyl)-ethyl moiety (Aaep) instead of the originally proposed cyclic guanidine [26].

### 2.3. Genus Nostoc

In 2013, Kapuścik and colleagues obtained aeruginosin 865 (Figure 5) from *Nostoc* sp. Lukešová 30/93 for the first time. The structure of aeruginosin 865 was determined by 1D- and 2D-NMR. It is the first aeruginosin-type polypeptide that contains both fatty acid and carbohydrate and the first aeruginosin that shows anti-inflammatory activity [11].

After a lapse of many years, varlaxin 1046A and varlaxin 1022A (Figure 6) were discovered from *Nostoc* sp. UHCC 0870 in 2022. Both of them were capable of inhibiting human trypsin isozymes at subnanomolar concentrations. The structure of the varlaxin variant was derived from 1D- and 2D-NMR data and the most significant difference between varlaxin and other aeruginosins is that the Choi moiety possesses two 4-hydroxyphenylacetic acid (Hpaa) modifications [12].

### 2.4. Genus Nodularia

In 1997, Fujii et al. first identified a new type of glycosylated aeruginosin, suomilide (Figure 6), in *Nodularia spumigena* (strain HKVV) [13]. The structure was elucidated by 2D-NMR combined with MS/MS technique. In 2021, Ahmed et al. also found suomilide in *Nodularia sphaerocarpa* (strain UHCC 0038) [14].

### 2.5. Sponges

In 2002, Quinn and colleagues isolated dysinosin A (Figure 7), a new type of aeruginosin, from a sponge in the family Dysideidae found near Lizard Island, North Queensland, Australia [15]. Dysinosin A is a potent inhibitor of the coagulation cascade factor VIIa2 and an inhibitor of the serine protease thrombin as well. The structure of dysinosin A was determined by using 2D-NMR combined with acid hydrolysis studies, and the dysinosin A-thrombin-hirugen (hirugen is N-acetylhirudin 53′-64′ with sulfato-Tyr63′ [32]) complex was analyzed by X-ray crystallography [19]. The configurations of chiral centers of dysinosin A were confirmed to be C5 (R), C12 (S), C14 (S), C15 (S), C17 (R) and C19 (S). The chemical exchange correlation between the high-intensity and low-intensity signals observed in the NOESY spectra suggests the presence of conformational isomers rather than structural isomers [15].

In 2003, Goetz and colleagues isolated chlorodysinosin A (Figure 7), a chloride derivative of dysinosin A (Figure 7), and characterized it with the same backbone structure and absolute configurations as that of dysinosin A [16]. Among the natural aeruginosin family, chlorodysinosin A is the most potent inhibitor of the serine proteases, thrombin, factor VIIa and factor Xa, which are key enzymes in the process leading to platelet aggregation and fibrin mesh formation in humans [25]. In the following year, Carroll and colleagues isolated three marine natural products, dysinosin B, C and D (Figure 7), from marine sponges in the family Dysideidae and determined their structures by 1D- and 2D-NMR [17]. Dysinosin D lacks the sulfate group compared to dysinosin A-C, and therefore its inhibition of factor VIIa and thrombin is enhanced by a factor of 10, suggesting that the sulfate group contributes to the binding of factor VIIa and thrombin [17]. Dysinosin B shares the same xylopyranose (Xyl) moiety as aeruginosin 205A and 205B (Figure 4) and thus belongs to the glycosylated aeruginosin.

## 3. Structural Diversity

Aeruginosins are highly variable linear tetrapeptides. The backbone of aeruginosin consists of four residues: a 4-hydroxyphenyl lactate derivative at the N-terminus [33], a hydrophobic amino acid [27], a Choi moiety and an arginine derivative at the C-terminus (Figure 8, Table 1) [29,34].

### 3.1. Diversity of N-Terminal Residue

The first position of the N-terminus of aeruginosin is usually occupied by the derivatives of hydroxyphenyl lactic acid (Hpla) or phenyllactic acid (Pla). Hpla is an NRPS compound that can be used for tyrosine metabolism. It is in the D-configuration in the vast majority of homologs, but it is in the L-configuration in aeruginosin KT608A (Figure 7) [27], and Pla is in the L-configuration in aeruginosin 205A (Figure 4) as well [9]. Hpla can be further modified by mono- or di-halogenation, hydroxylation and sulfation on the benzene ring [30]. In contrast, the Hpla of aeruginosin 98C (Figure 2), aeruginosin GE686, aeruginosin GE766, aeruginosin GE730 and aeruginosin GE810 (Figure 3) were found to be brominated, which is specific in comparison to other aeruginosins, as bromine cannot be detected in the natural environment or culture media [4,5]. More rarely, the first position of the N-terminus in the recently discovered suomilide, varlaxin 1046A and varlaxin 1022A (Figure 6) is occupied by 2-O-methylglyceric acid 3-O-sulfate (Mgs) [12,14].

### 3.2. Diversity of the Side Chain of the Second Residue

The second position of the N-terminus of aeruginosin is occupied by variable hydrophobic amino acids, relatively abundantly by leucine (Leu) and isoleucine (Ile), followed by phenylalanine (Phe), tyrosine (Tyr) and homotyrosine [35], while valine (Val) is the least abundant and so far only found in the dysinosin variant [15,17]. In most aeruginosins, these amino acids are in the D-configurations but are in the L-configurations in microcin SF608 (Figure 2) [6] and aeruginosin 205A (Figure 4) [9,30].

### 3.3. Modifications of Choi Moiety

Choi is one of the characteristics that distinguishes aeruginosin from other peptide compounds [35]. It has been confirmed in the organic synthesis of aeruginosin 298A (Figure 1) and 298B (Figure 2) that it is synthesized from tyrosine in vitro, but it is deduced to be biosynthesized from prephenate in vivo [36]. Choi is also highly variable and can be glycosylated, sulfated as well as halogenated at the R5 position of the general structural formula (Figure 8) [34]. The 5,6-OH in the Choi moiety of aeruginosin 865 (Figure 5) is replaced by ManA and HA, while suomilide (Figure 6) contains a multi-functional tricyclic azabicyclononane (Abn) moiety despite the absence of the Choi moiety. So suomilide still belongs to the aeruginosin family [11,14].

### 3.4. Diversity of the Side Chain of C-Terminal Residue

The C-terminal site is occupied by a variable arginine residue that presents a variety of types: the first is an argininol reside generated by the reduction of the carboxyl group of arginine to the hydroxyl group (aeruginosa 298-A) (Figure 1) [18]; the second is agmatine generated by decarboxylation of arginine (aeruginosin 98-A) (Figure 1) [22]; the third to the fourth is generated by the cyclization of the arginine side chain. Arginine was cycled to either a five-membered ring (Aaep: oscillarin) (Figure 5) [10] or a six-membered ring, and the six-membered ring presents in two forms: reduction of the carbonyl group to a hydroxyl group (aeruginosin 686-A) or the retention of the original carbonyl group (aeruginosin 686-B) (Figure 9) [30,31].

Aeruginosin 205A and 205B (Figure 4) are the most specific, independent of the above types, consisting of uncommon amino acids or amino acid derivatives, sugars and organic acids. At the time of their discovery, they were the only known glycosylate derivatives of aeruginosin [9,19]. Aeruginosin 205A and 205B (Figure 4) are specific glycopeptides, and their remarkable activity makes them potent candidates for drugs with protease inhibitory activity. Both of their backbones are composed of five residues: phenyllactic acid 2-O-sulfate (Pla), D-xyl, 3-hydroxyleucine (HLeu), 2-carboxy-6-chlorooctahydroindole (Ccoi) and agmatine. Among them, Pla, Ccoi and HLeu residues are rare in natural products. 3-hydroxyleucine has been found in peptide antibiotics such as telomycin, lysozyme peptides, and lactobacillin [37]. Aeruginosin 205A and 205B (Figure 4) have the same planar structure, but the stereochemistry of Hleu and Plas in aeruginosin 205B is opposite to that in 205A. Acid hydrolysis product data based on HPLC showed that the Pla residues of 205A and 205B were in the L- and D-configurations, respectively. The absolute stereochemistry of the Hleu residues was determined to be (2R,3S) and (2S,3R), respectively, using nine acid hydrolysates derived from Marfey’s reagent.

## 4. Biosynthetic Pathways

The distribution and functions of nonribosomal peptide synthases (NRPSs) and polyketide synthases (PKSs) have been extensively studied. They are two similar types of assembly machinery composed of multi-functional megaenzymes that are responsible for the synthesis of nonribosomal peptide (NRP) and polyketide (PK), respectively, in plants, bacteria and even fungi [38,39,40,41].

### 4.1. Polyketide Synthase (PKS)

PKs have a wide range of bioactivities due to their structural and functional diversity, but their biosynthetic mechanisms are similar. The generation of their core structures is catalyzed by PKS. PKS can be classified into three types based on their compositions, namely type I (modular), type II (iterative) and type III (chalcones) [42]. Type I PKS consists of modules with different core and auxiliary catalytic domains, including acyltrans-ferase (AT), acyl carrier protein (ACP), ketoacyl synthase (KS), ketoacyl reductase (KR), dehydratase (DH), enoylreductase (ER), methyltransferase (MT) and thioesterase (TE). AT, KS and ACP are the core functional domains for monomer assembly, and the PKS termination module ends with a thioesterase domain [43].

### 4.2. Nonribosomal Polypeptide Synthase (NRPS)

The backbone of NRP is synthesized by an assembly line consisting of multiple modules. A typical NRPS is composed of several modules in a certain order, typically 4 to 10 modules and some even 50 modules. Different domains with various enzymatic activities are responsible for assembling specific monomers to the nascent peptide chain. A standard module consists of three core domains, adenylation (A) domain, thiolation (T) domain/peptidyl carrier protein (PCP) domain and condensation (C) domain [39,44,45]. In addition to the C-A-T tri-domain, the NRPS module may also selectively contain some catalytic domains for offline modification, such as methylation (MT), oxidation (Ox), heterocyclization (Cy), epimerization (E) and sulfotransferase (ST) domains, etc. The C-terminus of NRPS usually contains a thioesterase (TE) or terminal condensation (CT) domain, which is responsible for releasing the products [46,47]. However, NRPSs that lack the TE domain or are replaced by an NAD (P) + dependent terminal reductase (R) domain may also exist, possibly by reductive release to terminate the peptide chain synthesis [48]. The vast majority of NRPSs follow the rules of collinear assembly. The number, type and arrangement order of NRPS modules are consistent with those of amino acid constituents of the product. In the process of biosynthesis, NRPS sequentially performs the catalytic function of each module to assemble monomers into specific NRPs according to a fixed logic. In addition, some NRPSs employ specific assembly mechanisms, such as module hopping, iterative extension and trans-uploading during product synthesis [49].

The iterative reaction process for the extension of the peptide chain of NRP mainly is: (1) the pantoyl–thioglyamine (Ppant) arm is tethered in the T/PCP domain (Figure 10A); (2) the A domain specifically recognizes the substrate amino acids to generate aminoacyl–AMP by consuming ATP, which activates the amino acid substrate [50,51], and the aminoacyl–AMP meets the pantoyl–thioglyamine (Ppant) arm tethered in the T/PCP domain and links to its free thiol group, resulting in the aminoacyl–S-carrier complex (Figure 10B) [52]; (3) the subsequent transfer of the amino-S-carrier complex from the A domain to the C domain, and the binding of the upstream carriers of aminoacyl, lipoyl CoA or peptidyl groups to form peptide bonds within the active site of the C domain (Figure 10C) [53]; (4) the TE domains typically present in the NRPS termination modules catalyze hydrolytic release or cyclization of the final products [54,55].

### 4.3. NRPS-PKS Hybrid

Both NRPS and PKS belong to a megasynthetase assembly line composed of multiple modules, using similar strategies for synthesis. Recently, some secondary microbial metabolites have been shown to require both NRPS and PKS domains to participate in the synthesis [56]. With the development of genome mining technology, it has been found that this pathway is ubiquitous in various microorganisms, such as the biosynthesis of aeruginosin [21,41].

In the initial step of aeruginosin biosynthesis, the modules responsible for the addition of α-keto acid include adenylation (A), ketoreductase (KR) and the peptidyl carrier protein (PCP) domains. The A domain uses a hitherto not fully revealed mechanism to specifically select α-keto acids, distinguishing them from α-amino acids and α-hydroxy acids: aspartic acid in contact with the α-amino acid in the amino acid selective A domain is substituted with a hydrophobic residue in the α-keto acid selective A domain [57]. The α-keto acid adenylated by this type of A domain is then transferred to the PCP domain. Subsequently, the PCP domain transfers the α-ketoacyl to the KR domain for stereoselective reduction of the keto group (Figure 11) [56]. Afterward, the α-hydroxyacyl-PCP is supplied to the C domain of the downstream module for condensation [56].

### 4.4. Biosynthesis of Aeruginosin

AerB, AerG, AerD, AerE, and AerF are found to be present in the biosynthesis pathway of all the aeruginosins discovered and perform similar functions, being responsible for peptide intermediate assembly and Choi precursor synthesis [14]. In the biosynthesis of most types of aeruginosins, AerA first activates and loads the substrate monocarboxylate; then, AerB catalyzes the addition of hydrophobic D-amino acid; biosynthesis of the third residue Choi is deduced to be initiated from prephenate and is catalyzed by non-NRPS enzymes, such as AerD, AerE, AerF [36,41,58,59] and AerK [30], in an offline manner and is supplied to AerG, which is a module responsible for adding Choi moiety to the elongating peptide [30]; for strains harboring the *aerM* gene, AerM is responsible for the C-terminal extension of aeruginosin, and the R domain of AerM is responsible for the formation of the C-terminal structure of aeruginosin and the release of the final product [60,61] (Figure 12). In contrast, the arginine residue of aeruginosin produced in the strain lacking *aerM* gene is synthesized by AerH to generate Aeap residue [41]. *aerO*, *aerP* and *aerQ* genes have so far only been found in the varlaxin biosynthesis gene cluster [12]. AerI and AerL are deduced to be responsible for the glycosylation and sulfation of Choi moiety [14,21], respectively. The details of the biosynthesis of several representative types of aeruginosins are listed below.

#### 4.4.1. Aeruginosin 126A

AerA, AerB, AerD, AerE, AerF, AerG, AerH and AerI mainly participate in the biosynthesis of aeruginosin 126A [41] (Figure 13A). AerA is a PKS-like module containing A, KR, and ACP domains. AerA activates and tethers phenylpyruvate, which is then reduced by the KR domain to generate phenyl lactic acid (Plac) moiety. AerB contains the C, A, PCP, and E domains. Since the sequence of the substrate binding pocket of the A domain is very similar to that of the leucine activation domain of McyB involved in microcystin biosynthesis, AerB is deduced to be responsible for the addition of leucine to the peptide chain. AerD, E, and F are involved in Choi biosynthesis. AerG is a dimodule NRPS with domain order C-A-PCP-C-PCP. The substrate binding pocket of the A domain of AerG is shown to be most similar to enzymes that activate proline or methylproline, but it activates a proline-like amino acid, Choi. The second module of AerG is probably responsible for the incorporation of C-terminal residue. AerH shares similarities with a variety of bacterial oxygenases, which may function in the synthesis of Aaep from arginine or agmatine. AerI possesses sequence similarity to glycosyltransferases, so xylose moiety is postulated to be transferred to the hydroxyl group of Choi by AerI.

#### 4.4.2. Aeruginosin 686A

AerA, AerB, AerD, AerE, AerF, AerK AerG and AerM mainly participate in the biosynthesis of aeruginosin 686A [30] (Figure 13B). AerA is a hybrid NRPS/PKS module that includes A, KR and T domains. The A domain activates the substrate hydroxyphenylpyruvate, which is anchored by the T domain and reduced to HPla by the KR domain and is further halogenated by AerJ. AerB is responsible for the addition of tyrosine to the elongating peptide chain. AerD, AerE, AerF and AerK are involved in the formation of Choi. AerG consists of three core domains, C, A, and T, to load and incorporate Choi moiety into the elongating peptide. AerM, consisting of C, A, T and R domains, is responsible for the C-terminal extension. The R domain of AerM catalyzes the formation of the structure of the C-terminal arginine residue and hydrolysis of the thioester bond to release the assembled peptide chain.

#### 4.4.3. Dysinosin B

AerB, AerD, AerE, AerF, AerG and AerI mainly participate in the biosynthesis of dysinosin B [14] (Figure 13C). AerB is a multimodular NRPS megasynthetase responsible for glycerate loading and sulfation modification, as well as the addition of enantiomerized valine. AerD, AerE and AerF are involved in the synthesis of the Choi moiety. The two modules of AerG are responsible for the loading of Choi and arginine, respectively. Similar to the case in aeruginosin 126A biosynthesis, AerI modifies the hydroxyl group of Choi by glycolysation.

#### 4.4.4. Aeruginosin NAL2

AerB, AerD, AerE, AerF, AerG and AerM mainly participate in the biosynthesis of aeruginosin NAL2, which is predicted to be initiated by loading a short-chain fatty acid via C domain of AerB [21] (Figure 13D). The tethered short-chain fatty acid is then linked to tyrosine in AerB. AerD, AerE and AerF participate in the synthesis of Choi, which is submitted to AerG for adding to the elongating peptide chain. AerM is responsible for recognizing the substrate arginine to generate agmatine and its incorporation in the peptide. Despite the presence of the AerI-encoding gene in the genome of the producing strain, no glycosylation modification was found in aeruginosin NAL2.

#### 4.4.5. Aeruginosin 865

The biosynthetic gene cluster (BGC) of aeruginosin 865 was inferred from the BGC of nostopeptolide A1 to contain homologs of *aerA*, *aerB*, *aerD*, *aerE*, *aerF*, *aerG* and *aerN* in the general BGC [62] (Figure 13E). Genes encoding glycoside modifying enzymes and an enzyme predicted to be an acyltransferase are also present in aeruginosin 865 BGC. This leads to chemical differences between aeruginosin 865 and other analogs of aeruginosin in the presence or absence of glycoside and hexacarbon fatty acid tail.

#### 4.4.6. Suomilide

AerB, AerD, AerE, AerF, AerK, AerG AerI and AerH mainly participate in the biosynthesis of suomilide [14] (Figure 13F). The FkbH domain in the first module of AerB is probably responsible for loading glycerate, which is further methylated and sulfonated by MT and ST domains, respectively [63]. The second module of AerB activates L-isoleucine, which is incorporated into the elongating peptide chain. AerD, AerE, AerF, AerK and AerH are deduced to be responsible for the synthesis of Abn moiety converted from Choi by an unknown mechanism. Abn is added to the elongating peptide by the first module in AerG, while the second module of AerG is responsible for the loading of arginine. AerI and a membrane-bound O-acyl transferase (MBOAT) enzyme are deduced to catalyze the glycosylation and further acylation of Abn, respectively. Finally, AerH converts arginine to Aaep by a currently unknown mechanism.

#### 4.4.7. Varlaxin

AerB, AerD, AerE, AerF, AerG, AerP and AerQ mainly participate in the biosynthesis of varlaxin [12] (Figure 13G). Both AerB and AerG are two bimodular NRPS enzymes that are responsible for the formation of the backbone of varlaxin. The first module of AerB contains the O-methylglyceric acid transferase (OMT), FkbH, PCP and ST domains for glycerate incorporation and further methylation and sulfation of this residue. The second module of AerB loads isoleucine, which is converted to the D-configuration from the L-configuration via the E domain. AerD, AerE and AerF are responsible for the synthesis of Choi. AerG consists of two modules for loading Choi and arginine, respectively. AerO contains an A domain responsible for the recognition of Hpaa, while AerP contains a PCP domain that loads Hpaa. AerQ belongs to the MBOAT enzyme family that is responsible for the glycosylation of Hpaa.

## 5. Bioactivity

Proteases play important roles in numerous important biological processes, from simple proteolysis to the degradation of important regulators of major cellular pathways. Aeruginosin is a chemically diverse family of serine protease inhibitors, and its inhibitory activity is largely related to C-terminal modifications [19]. It demonstrates a high degree of inhibition of thrombin and trypsin in vitro (Table 1). Moreover, the aeruginosins with C-terminal argininal residues show a more significant tendency to inhibit thrombin than the aeruginosin with C-terminal agmatine or argininol [4].

### 5.1. Thrombin Inhibitory Activity

Cardiovascular disease (CVD) is not only affected by external environmental factors but is also closely related to metabolism status [64]. The coagulation system and its components have a direct impact on CVD [65]. Blood coagulation is a process consisting of a series of complicated chain reactions. As the last enzyme participating in the coagulation system, thrombin plays a central role in the process of hemostasis, inducing platelet aggregation and secretion [66,67]. Thrombin is involved in many biochemical reactions, the most important function of which is the conversion of cleaved fibrinogen into fibrin. Fibrin is subsequently converted into a cross-linked network that forms a thrombus with bound platelets [32]. Over the past few decades, breakthroughs in antithrombotic drugs have been made, but they have been limited by side effects and poorly targeted effects [66]. Efforts have been made to discover novel antithrombotic drugs that can specifically and directly inhibit thrombin [67,68].

Few low molecular-weight natural product is a selective inhibitor of thrombin at present. Because of the strong inhibition of coagulation factors exhibited by aeruginosin, it has become a key candidate in the development of anticoagulants. To date, nearly 100 compounds of this family have been isolated, many of which are thrombin inhibitors [12]. The binding pattern of aeruginosa 298-A (Figure 1) in thrombin is similar to that of other serine protease inhibitors: it binds to the active site of thrombin in a non-covalent manner [69]. Oscillarin (Figure 5) has an inhibitory concentration of 0.02 μM on thrombin, which is one of the most effective thrombin inhibitors in the aeruginosin family [26]. Dysinosin A (Figure 7) is an inhibitor of factor VIIa and thrombin with Ki values of 0.11 μM and 0.45 μM, respectively. Compared with dysinosin A-D (Figure 7) showed reduced thrombin activity, glycosylated dysinosin B (Figure 7) was a more potent inhibitor of factor VIIa with a Ki value of 0.09 μM [17].

### 5.2. Trypsin Inhibitory Activity

Trypsin is an enzyme that plays a major role in the digestion of food but also has important functions beyond that in the digestive system [14]. Cancer develops as a gradual transformation of normal cells into highly malignant cells, and advanced stages of cancer are often difficult to be treated [12]. Proteases play a crucial role in the metastatic spread of cancer cells and tumor growth, and trypsin is one of the most characteristic protein hydrolases [70]. Trypsin-1, -2 and -3 are three isozymes from human with highly similar structures and functions. Trypsin-3 is demonstrated to be capable of promoting tumor growth and metastasis in several types of cancer, including prostate, breast and pancreatic cancers. Therefore, trypsin-3 has also been considered a potential target for the treatment of these cancers [71,72].

A number of members of the aeruginosin family exhibit strong inhibition of trypsin at low micromolar to low nanomolar concentrations. Since the isolation of aeruginosin 298A (Figure 1) in 1994 [18], most of the reported aeruginosins have exhibited trypsin-inhibitory activity [12,14,19,33]. However, aeruginosin 298B (Figure 2) [19] and aeruginosin EI461 [73] did not exhibit any trypsin inhibitory activity due to the absence of a C-terminal arginine derivative [5]. Most biochemical assays of aeruginosins only use porcine and bovine trypsins as inhibitory targets, whereas the sequence of human trypsin is significantly different from that of porcine and bovine trypsins, which biases the preclinical evaluation of aeruginosins. Suomilide (Figure 6), discovered in *Nodularia spumigena* HKVV in 1997, has been reported to inhibit human trypsin at low micromolar concentrations (Table 1). Moreover, suomilide inhibits human trypsin-1 to a lesser extent compared to human trypsin-2 and -3. The results of Ahmed’s study showed that suomilide could inhibit metastasis of prostate cancer cells [14]. In the following year, Heinilä and colleagues isolated varlaxin1046A and 1022A (Figure 6) from *Nostoc* sp. UHCC 0870 [12]. These two varlaxin variants exhibited strong inhibitory activity against porcine trypsin, and they were tested for inhibition of the three human trypsin isoenzymes. Varlaxin showed a similar inhibition profile as suomilide. Varlaxin 1046A showed approximately 50 to 200 times greater inhibitory activity against the trypsin isoenzyme than that of varlaxin 1022A. The only difference between these two varlaxins is the fourth residue: Aaep in varlaxin 1046A and Agma in varlaxin 1022A (Figure 6). Thus, aeruginosin is expected to be a pioneering molecule for the drug development of trypsin inhibitors.

### 5.3. Other Bioactivities

Plasmin is a hydrolytic enzyme that specifically degrades fibrin gel. It is produced by the proteolytic cleavage of blood plasminogen in humans. Under normal conditions, the anticoagulant and fibrinolytic systems of the coagulation system are in balance [74]. An imbalance in this process can lead to coagulation or hemorrhage, depending on which direction is dominant [75]. Therefore, fibrinolytic inhibitors are indispensable in the fibrinolytic system to regulate the balance of the fibrinolytic system. In addition, researchers have found that fibrinolytic enzymes are associated with the invasion and metastasis of cancer cells, so these enzymes are expected to be important candidates for anticancer targets in the future [76]. Most of the aeruginosins have been reported to be inhibitory to thrombin and trypsin, but some have also shown inhibition of fibrinolytic enzymes [19]. To date, the most potent inhibitor of plasmin in the aeruginosin family was aeruginosin 89A (Figure 2), IC_50_ of which reached 0.02 μM (Table 1) [4].

Inflammation underlies the pathogenesis of many serious diseases, such as CVD and Alzheimer’s disease, and also increases the risk of cancer development [77]. Aeruginosin 865 (Figure 5) exhibits not only inhibitory activity against trypsin but also shows anti-inflammatory activity that is absent in general aeruginosins. Interleukin-8 (IL-8) is a cytokine of the chemokine family. Its major function is to attract and activate neutrophils to play a role in inflammatory sites so as to achieve the goal of bactericidal [78]. In addition, IL-8 is also a potent angiogenic promoter [79]. Macrophage antigen-1 (Mac-1) is a transmembrane glycoprotein responsible for the translocation of leukocytes through the endothelium to inflammatory tissues [80]. In the in vitro AlphaLISA assay of IL-8 [81] and intercellular adhesion molecule-1 (ICAM-1) [82] on human pulmonary microvascular endothelial cells (HLMVECs), Kapuścik and colleagues treated HLMVECs with different concentrations of aeruginosin 865 (Figure 5) before stimulation with HTNF-a. With the increase in the concentration of aeruginosin 865, IL-8 and ICAM-1 were significantly down-regulated, which indicated that aeruginosin 865 had a high anti-inflammatory effect [11]. In addition, the observations of impedance measurements by using electrical cell-substrate impedance sensing (ECIS) showed that after the treatment of aeruginosin 865, the cell membranes of HLMVECs were intact and the cells were not suffered from cytotoxic [11]. Taken together, aeruginosin 865 is an immunomodulatory agent with significant anti-inflammatory activity and no cytotoxicity, which is consistent with the new demand for the treatment of immune diseases in the future.

Despite efforts to find an effective anticoagulant to replace existing heparin or warfarin therapy, it has been difficult to find a small molecule agent that is effective, safe and orally available; similar cases exist in the treatment of cancer metastasis as well. Aeruginosins, a family of newly discovered naturally occurring serine protease inhibitors, exhibit good inhibitory effects on trypsin, thrombin and plasmin, as summarized in Table 1. Therefore, these peptides are promising in the development of therapeutic agents for anti-thrombosis and cancer prognosis. However, they have not been applied to clinical trials and are only considered lead compounds at present. The reason for this fact may be relevant to the structural complexity of aeruginosins and inadequate investigation of the biosynthetic mechanism of aeruginosins, as illustrated above, hindering the accurate examination of structure–activity correlations and engineering of these peptides.

## 6. Conclusions and Prospects

Throughout history, natural products have been a valuable source of new molecular frameworks with diverse bioactivities. Since its first isolation in 1994, aeruginosin has attracted much attention from biologists, chemists and pharmacologists because of its special Choi structure and serine protease inhibitory activity, and it has been considered a promising drug candidate. As an important secondary metabolite rich in cyanobacteria and sponges, aeruginosins warrant an in-depth study of the relationship between their structure and bioactivity. Although a number of natural members of the aeruginosin family have been isolated to date, more compounds are needed to more meticulously delineate structure–activity relationships (SARs) and identify important structural motifs for various biological activities.

During a long evolutionary process, microorganisms have relied on linear combinations of various types of domains to obtain thousands of NRPSs, thus creating a diversity of structures and functions of NRP natural products. To date, even though many members of the aeruginosin family have been isolated, more compounds are still needed to deepen the understanding of the NRPS-PKS synthetic pathway [83]. The modular structural features of NRPS offer the possibility to artificially design and engineer NRP assembly lines and biosynthesize NRPs with novel backbones that can be used for drug screening. The complexity of their structures poses a great challenge to their synthesis. Research in this field is focused on the artificial modification of the NRPS synthesis mechanism after clarifying the biosynthetic pathway and the function of each domain in order to allow the biosynthetic production of a wide range of artificial aeruginosins, which can lead to enhanced bioactivity or a wider range of applications. At present, adenylation domain-specific rearrangements [84,85,86], multiple domain substitutions [87] and docking domain modifications [88,89] are the dominant approaches.

The synthesis of aeruginosin has greatly facilitated the creation of novel analogs applied to the healthcare of humans, which is the focus of future research. As a serine protease inhibitor, future perspectives of aeruginosin studies should focus more on finding better activity as well as solving the structural and pharmacological aspects of these compounds to enable better efficacy. At the same time, the sustainable production or engineering application of aeruginosin drugs through the heterologous expression of genes involved in the aeruginosin biosynthesis pathway will be a promising alternative for future chemical de novo synthesis. It is hoped that safe and effective drugs can be synthesized in this field in the future to treat common life-threatening diseases such as thrombosis and cancer.

Altogether, we have provided a detailed and comprehensive overview of the studies of aeruginosins in terms of biogenesis, structural diversity, biosynthesis and multiple bioactivities, which pave the way for the preclinical trials of these highly diverse nonribosomal linear tetrapeptides probably carried out in the future.

## Figures and Tables

**Figure 1 marinedrugs-21-00217-f001:**
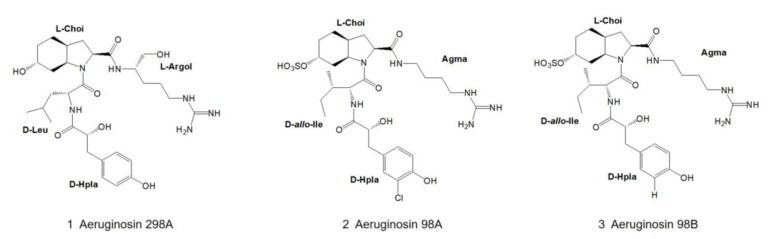
Structures of aeruginosin 298A, 98A and 98B.

**Figure 2 marinedrugs-21-00217-f002:**
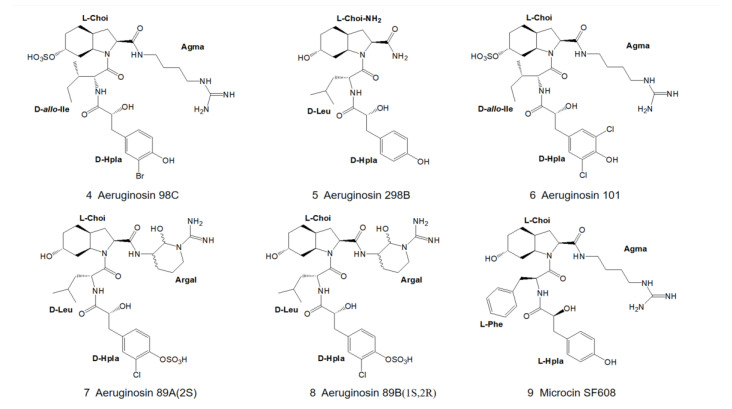
Structures of aeruginosin 98C, 298B, 101, 89A, 89B, and microcin SF608.

**Figure 3 marinedrugs-21-00217-f003:**
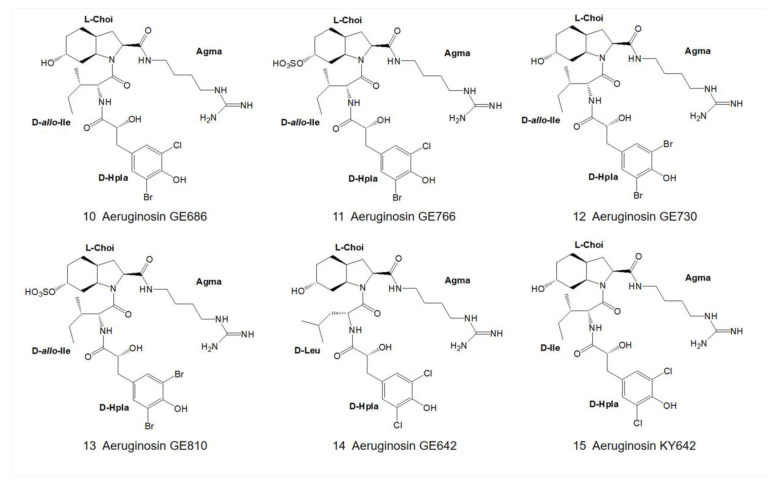
Structures of aeruginosin GE686, GE766, GE730, GE810, GE642 and KY642.

**Figure 4 marinedrugs-21-00217-f004:**
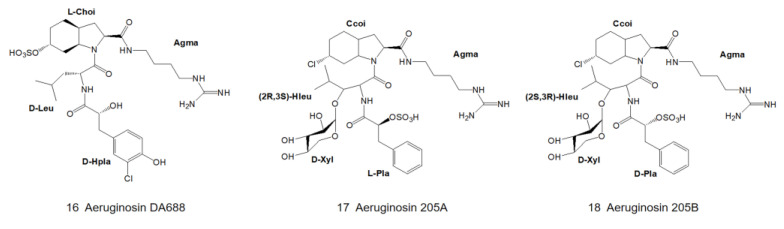
Structures of aeruginosin DA688, 205A and 205B.

**Figure 5 marinedrugs-21-00217-f005:**
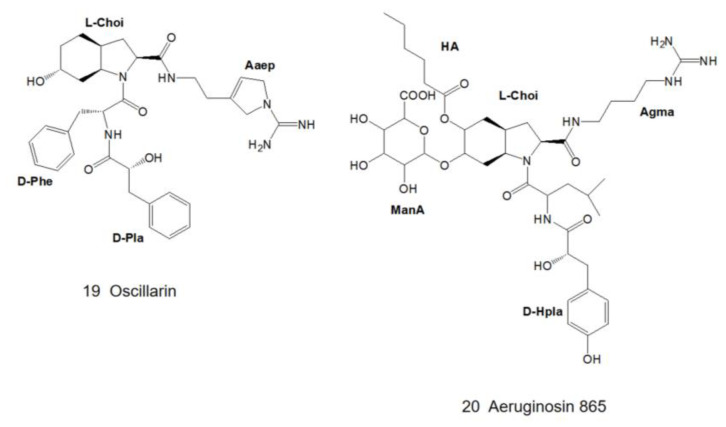
Structures of oscillarin and aeruginosin 865.

**Figure 6 marinedrugs-21-00217-f006:**
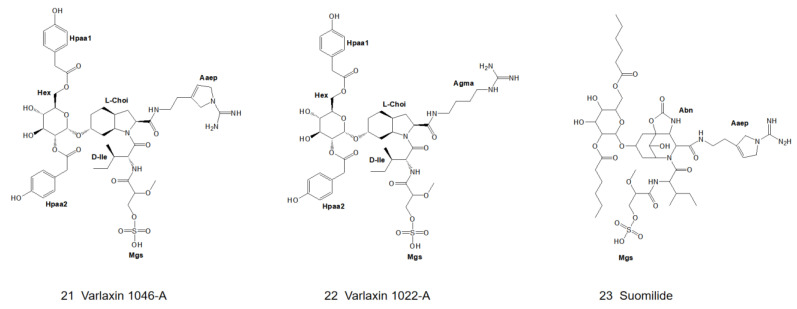
Structures of varlaxin 1046-A, 1022-A, and suomilide.

**Figure 7 marinedrugs-21-00217-f007:**
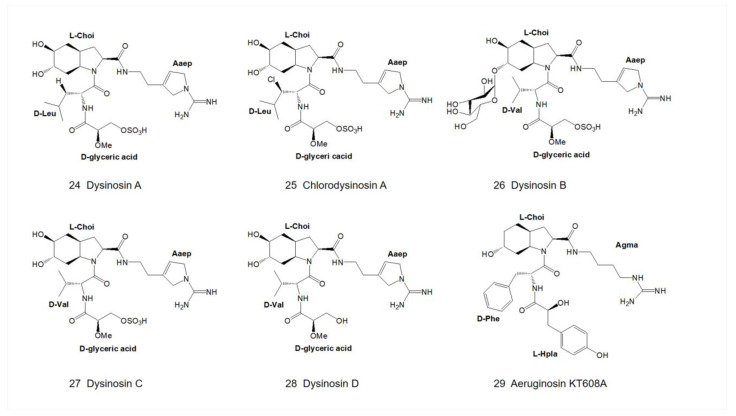
Structures of dysinosin A-D, chlorodysinosin A and aeruginosin KT608A.

**Figure 8 marinedrugs-21-00217-f008:**
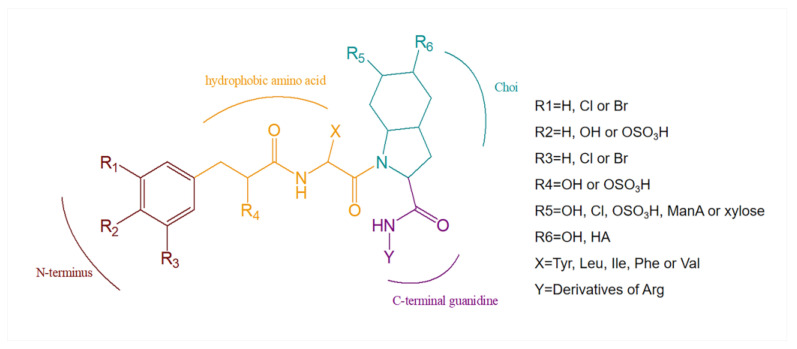
The general structure of aeruginosin.

**Figure 9 marinedrugs-21-00217-f009:**
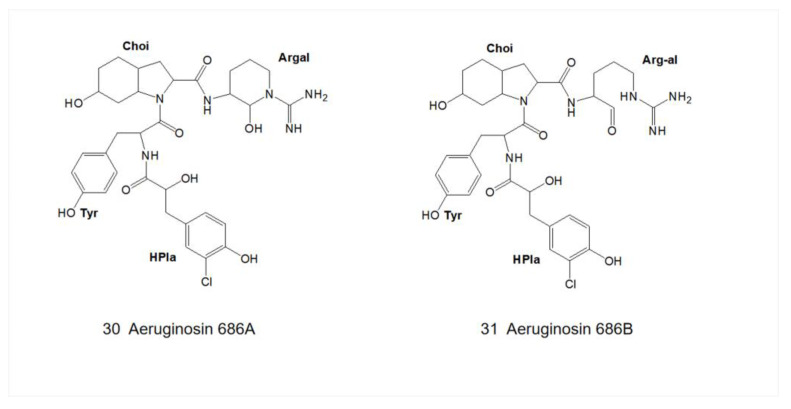
Structures of aeruginosin 686A and 686B.

**Figure 10 marinedrugs-21-00217-f010:**
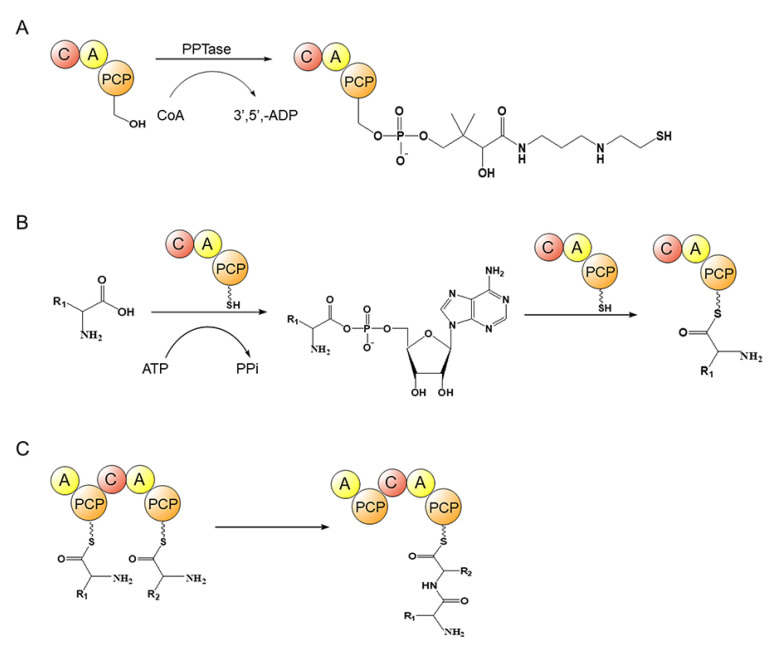
A canonical peptide chain extension process during biosynthesis of aeruginosin. (**A**) Pantoyl–thioglyamine (Ppant) arm is tethered in the PCP domain, which is catalyzed by PPTase; (**B**) amino acid substrate is activated by A domain and is loaded onto the PCP domain, resulting in the aminoacyl–S-carrier complex; (**C**) peptide bond formation catalyzed by C domain. PPTase: 4′-phosphopantetheinyl transferase; A: adenylation domain; C: condensation domain; PCP: peptidyl carrier protein domain; CoA: coenzyme A; 3′,5′-ADP: adenosine 3′,5′-diphosphate; ATP: adenosine triphosphate; PPi: pyrophosphoric acid.

**Figure 11 marinedrugs-21-00217-f011:**
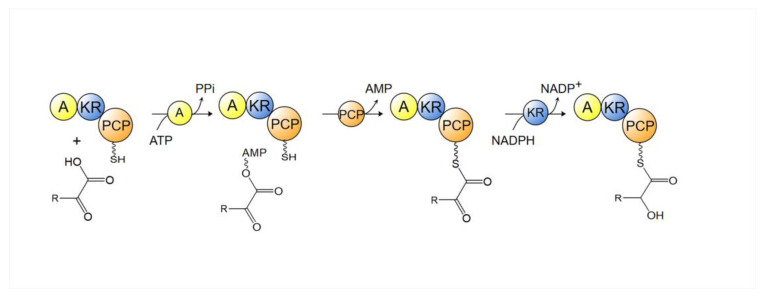
The reaction process of the loading of α-keto acid. A: adenylation domain; KR: ketoreductase domain; PCP: peptidyl carrier protein domain; NADPH: nicotinamide adenine dinucleotide phosphate.

**Figure 12 marinedrugs-21-00217-f012:**
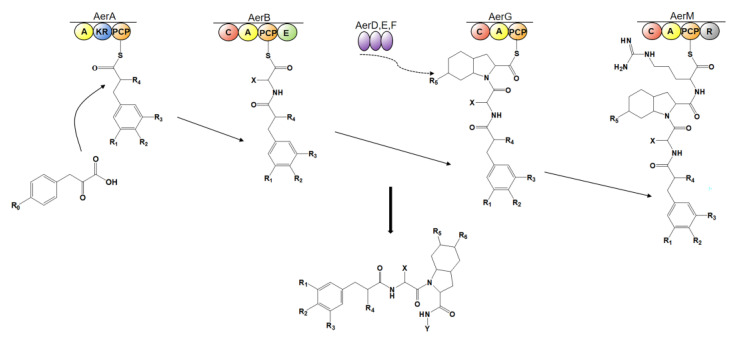
A general biosynthetic scheme of aeruginosin. A: adenylation domain; KR: ketoreductase domain; C: condensation domain; PCP: peptidyl carrier protein domain; E: epimerization domain; R: reductase domain.

**Figure 13 marinedrugs-21-00217-f013:**
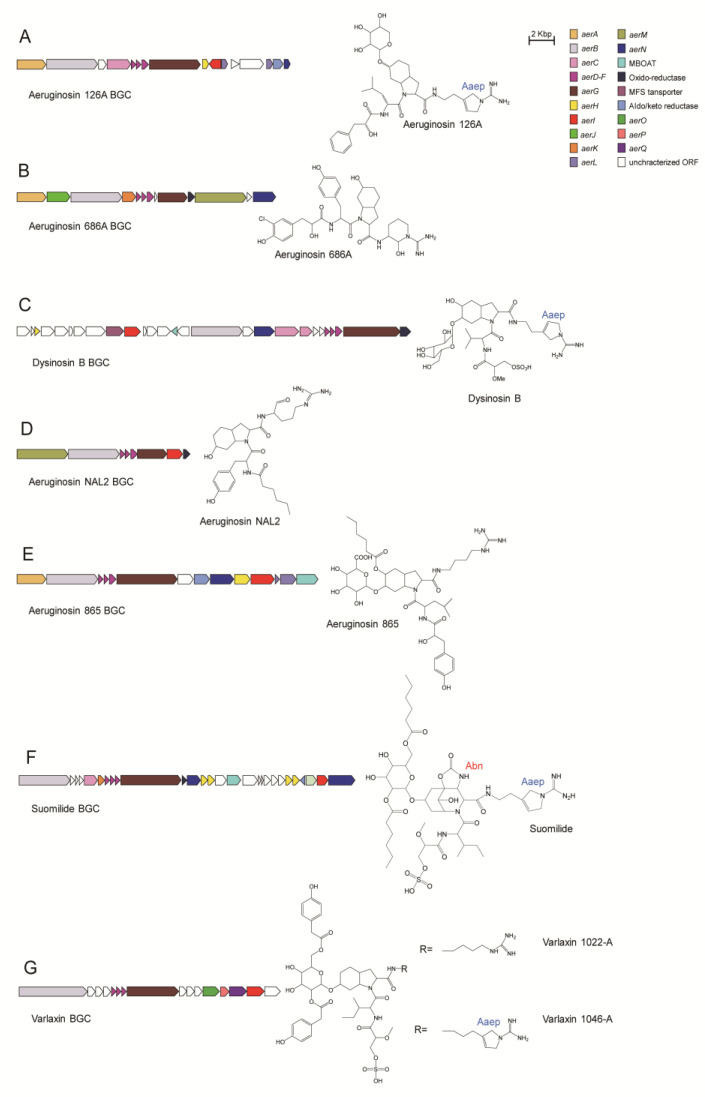
Biosynthetic gene clusters (BGCs) and structures of several representative types of aeruginosins: aeruginosin 126A (**A**), aeruginosin 686A (**B**), dysinosin B (**C**), aeruginosin NAL2 (**D**), aeruginosin 865 (**E**), suomilide (**F**), varlaxin 1022A and 1046A (**G**).

**Table 1 marinedrugs-21-00217-t001:** Biogenesis, structural diversity, and bioactivity of aeruginosins. Hpla: 4-hydroxyphenyllactic acid; Pla: phenyllactic acid; Choi: 2-carboxy-6-octahydroindole; Abn: tricyclic azabicyclononane; Agma: 4-amidinobutylamide; Aaep: 1-amidino-3-(2-aminoethyl)-3-pyrroline; Mgs: 2-O-methylglyceric acid 3-O-sulfate.

No.	Name	Source	Structure Diversity				IC_50_			Refs.
			The N Terminal Residue	The Second Residue	The Third Residue	The C Terminal Residue	Trypsin	Thrombin	Plasmin	
1	Aeruginosin 298A	*Microcystis aeruginosa* NIES-298	R_2_=OH, R_4_=OH D-Hpla	X = side chain of D-Leu	R_5_=OH L-Choi	L-Argol	1.38 μM	0.42 μM	>14 μM	[4,28]
2	Aeruginosin 98A	*Microcystis aeruginosa* NIES-98	R_1_=Cl, R_2_=OH, R_4_=OH D-Hpla	X = side chain of D-*allo*-Ile	R_5_=OSO_3_H L-Choi	Agma	0.87 μM	10.17 μM	8.72 μM	[20,29]
3	Aeruginosin 98B	*Microcystis aeruginosa* NIES-98	R_1_=H, R_2_=OH, R_4_=OH D-Hpla	X = side chain of D-*allo*-Ile	R_5_=OSO_3_H L-Choi	Agma	0.92 μM	15.28 μM	10.7 μM	[20,29]
4	Aeruginosin 98-C	*Microcystis aeruginosa* NIES-98	R_1_=Br, R_2_=OH, R_4_=OH D-Hpla	X = side chain of D-*allo*-Ile	R_5_=OSO_3_H L-Choi	Agma	5.33 μM	4.5 μM	6.83 μM	[29]
5	Aeruginosin 298B	*Microcystis aeruginosa* NIES-298	R_2_=OH, R_4_=OH D-Hpla	X = side chain of D-Leu	R_5_=OH L-Choi-NH_2_	-	>100 μM	>100 μM	>100 μM	[4]
6	Aeruginosin 101	*Microcystis aeruginosa* NIES-101	R_1_=Cl, R_2_=OH, R_3_=Cl, R_4_=OH D-Hpla	X = side chain of D-*allo*-Ile	R_5_=OSO_3_H L-Choi	Agma	4.15 μM	4.43 μM	4.57 μM	[4]
7	Aeruginosin 89A	*Microcystis aeruginosa* NIES-89	R_1_=Cl, R_2_=OSO_3_H, R_4_=OH D-Hpla	X = side chain of D-Leu	R_5_=OH L-Choi	L-Argal	0.48 μM	0.04 μM	0.02 μM	[4]
8	Aeruginosin 89B	*Microcystis aeruginosa* NIES-89	R_1_=Cl, R_2_=OSO_3_H, R_4_=OH D-Hpla	X = side chain of D-Leu	R_5_=OH L-Choi	D-Argal	7.9 μM	0.06 μM	0.55 μM	[4]
9	Microcin SF608	*Microcystis aeruginosa*	R_2_=OH, R_4_=OH L-Hpla	X = side chain of L-Phe	R_5_=OH L-Choi	Agma	0.82 μM	-	-	[29]
10	Aeruginosin GE686	*Microcystis aeruginosa* from bloom material	R_1_=Br, R_2_=OH, R_3_=Cl, R_4_=OH D-Hpla	X = side chain of D-*allo*-Ile	R_5_=OH L-Choi	Agma	3.2 μM	12.8 μM	-	[5]
11	Aeruginosin GE766	*Microcystis aeruginosa* from bloom material	R_1_=Br, R_2_=OH, R_3_=Cl, R_4_=OH D-Hpla	X = side chain of D-*allo*-Ile	R_5_=OSO_3_H L-Choi	Agma	12.2 μM	>45.5 μM	-	[5]
12	Aeruginosin GE730	*Microcystis aeruginosa* from bloom material	R_1_=Br, R_2_=OH, R_3_=Br, R_4_=OH D-Hpla	X = side chain of D-*allo*-Ile	R_5_=OH L-Choi	Agma	2.3 μM	12.9 μM	-	[5]
13	Aeruginosin GE810	*Microcystis aeruginosa* from bloom material	R_1_=Br, R_2_=OH, R_3_=Br, R_4_=OH D-Hpla	X = side chain of D-*allo*-Ile	R_5_=OSO_3_H L-Choi	Agma	18.2 μM	>45.5 μM	-	[5]
14	Aeruginosin GE642	*Microcystis aeruginosa* from bloom material	R_1_=Cl, R_2_=OH, R_3_=Cl, R_4_=OH D-Hpla	X = side chain of D-Leu	R_5_=OH L-Choi	Agma	8.5 μM	>45.5 μM	-	[5]
15	Aeruginosin KY642	*Microcystis aeruginosa* from bloom material	R_1_=Cl, R_2_=OH, R_3_=Cl, R_4_=OH D-Hpla	X = side chain of D-Ile	R_5_=OH L-Choi	Agma	1.85 μM	-	-	[5,7]
16	Aeruginosin DA688	*Microcystis aeruginosa* from bloom material	R_1_=Cl, R_2_=OH, R_3_=H, R_4_=OH D-Hpla	X = side chain of D-Leu	R_5_=OSO_3_H L-Choi	Agma	9.5 μM	>45.5 μM	-	[8]
17	Aeruginosin 205A	*Planktothrix agardhii* NIES-205	R_4_=OSO_3_H L-Pla	X = side chain of (2R,3S)-Hleu	R_5_=Cl Ccoi	Agma	0.08 μM	1.65 μM	-	[4,9]
18	Aeruginosin 205B	*Planktothrix agardhii* NIES-205	R_4_=OSO_3_H D-Pla	X = side chain of (2S,3R)-Hleu	R_5_=Cl Ccoi	Agma	0.08 μM	0.19 μM	-	[4,9]
19	Oscillarin	*Planktothrix agardhii* B2 83	R_4_=OH, D-Pla	X = side chain of D-Phe	R_5_=OH L-Choi	Aaep	0.03 μM	0.02 μM	>300 μM	[10,26]
20	Aeruginosin 865	*Nostoc* sp. Lukešová 30/93	R_2_=OH, R_4_=OH D-Hpla	X = side chain of D-Leu	R_5_=ManA,R_6_=HA Choi	Agma	-	-	-	[11]
21	Varlaxin 1046A	*Nostoc* sp. UHCC 0870	Mgs	X = side chain of D-Ile	Hex	Aaep	0.62–3.6 nM	-	-	[12]
22	Varlaxin 1022A	*Nostoc* sp. UHCC 0870	Mgs	X = side chain of D-Ile	Hex	Agma	97–230 nM	-	-	[12]
23	Suomilide	*Nodularia sphaerocarpa* UHCC 0038	Mgs	X = side chain of *allo*-Ile	Abn	Aaep	1.8 μM	-	-	[12,13]
24	Dysinosin A	Species in the family Dysideidae	R4=H D-glyceric acid	X = side chain of D-Leu	R5=OH R6=OH L-Choi	Aaep	-	0.38 μM	-	[15,19]
25	Chlorodysinosin A	-	D-glyceric acid	X = side chain of D-Leu	R_5_=OH L-Choi	Aaep	0.03 μM	0.004 μM	-	[19,25]
26	Dysinosin B	*Lamellodysidea chlorea*	D-glyceric acid	X = side chain of D-Val	R_5_=xylose,R_6_=OH L-Choi	Aaep	-	0.17 μM	-	[17]
27	Dysinosin C	*Lamellodysidea chlorea*	D-glyceric acid	X = side chain of D-Val	R_5_=OH,R_6_=OH L-Choi	Aaep	-	0.55 μM	-	[17]
28	Dysinosin D	*Lamellodysidea chlorea*	D-glyceric acid	X = side chain of D-Val	R_5_=OH,R_6_=OH L-Choi	Aaep	-	>5.1 μM	-	[17]
29	Aeruginosin KT608A	*Microcystis aeruginosa* from bloom material	R_2_=OH, R_4_=OH L-Hpla	X = side chain of D-Phe	R_5_=OH L-Choi	Agma	1.9 μM	-	-	[12,27]
30	Aeruginosin 686A	*Microcystis aeruginosa* PCC 7806	R1=Cl, R2=OH, R4=OH	X = side chain of D-Tyr	R_5_=OH L-Choi	Argal	-	-	-	[30,31]
31	Aeruginosin 686B	*Microcystis aeruginosa* PCC 7806	R1=Cl, R2=OH, R4=OH	X = side chain of D-Tyr	R_5_=OH L-Choi	Arg	-	-	-	[30,31]

## Data Availability

No new data were created or analyzed in this study. Data sharing is not applicable to this article.

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
