# Peer review of "Diversity, Biosynthesis and Bioactivity of Aeruginosins, a Family of Cyanobacteria-Derived Nonribosomal Linear Tetrapeptides"

_marinedrugs, 2023, doi:10.3390/md21040217_

Round 1
Reviewer 1 Report
In general I enjoyed reading this MS, it is very impressive to read about structural variation within this exciting non-ribosomal synthesized peptide family. However, my general impression is that the authors have difficulties to set the scope and want to cover all three aspects, structural diversity, biosynthesis and bioactivity, which is possibly too much for this review. Furthermore, no aims are proposed (although to summarize the current knowledge might be already such an objective).
Another general impression is that – perhaps because of this impressive structural diversity – the review on structural diversity is not complete, and even entire peptide groups have been left out (including their organisms, i.e. Nodularia producing spumigins?).
The biosynthesis pathway of aeruginosin and its modifications is the weakest part of this MS and would need thorough revision. Indeed biosynthesis was first described from P. agardhii strain CYA126/8, and authors should start from there. In the past some enzymatic steps have been elucidated in more detail and analogies to other NRPS have been made. I cannot follow the logic of Figure 3 (which is more or less showing the general peptide chain elongation process). Much more important: What is the relationship between the various published gene clusters and observed structural modification?
The last part is on the bioactivity and potential pharmaceutical application. Again very interesting, but hard to see, what is the authors’ aim or conclusion (in addition to deepen our understanding). Consequently I would need to recommend major revision including a considerable reshaping of the review.
Specific comments
Title: use plural, i.e. aeruginosins, terapeptides
l.22: spelling error: Furthermore
l.41: Aeruginosins cannot be considered hepatotoxins neither toxins (from human point of view)
l.43: throughout the MS I found it rather confusing, that Refs are not ordered according to their appearance in the text, sometimes it seems wrongly cited, I.e. l.52, ref 27 is not on crystallographic structure of aeruginosin-protease complex?, or l.60, ref 27 is not on Oscillarin?
In addition, literature that is cited must be relevant, and well selected, please put quality of references before quantity, in numerous cases I have the impression that not the most relevant refs have been chosen.
l.92: spelling error: aeruginosin 298A, I think you mean ref. 18 (not ref. 17)
l.103: The organisms should be named according to modern taxonomy, eg. Oscillatoria agardhii NIES-205 is now assigned to Planktothrix agardhii NIES-205 (eg. Suda et al. 2002),
please realized that from Planktothrix agardhii strain CYA126/8 aeruginosins 126A+B were described in 2007 (Ishida et al. 2007)
l123: check sentence
l140: the ref. 69 must be wrong, I think you mean Heinilä et al. 2022
l129: the selection of producer organisms must be clear, i.e. why is Nodularia (spumigins) not mentioned?
L203: suomilide has been described already in 1997 (Fujii et al. 1997)
L226: units for IC50 given in Table 1 must be the same
L236: I think the relevant review is Marahiel et al. 1997, Chem. Rev.
L241: If I remember rioght, the first biosynthesis pathway was decribed from P. agardhii CYA126/8 (Ishida et al. 2007) including an aeruginosin synthesis gene inactivation mutant
L254: check “adenylation” in PKS? (see also legend in Fig. 2)
L287: I think you mean “colinear assembly”
L311: Figure 3, fo which purpose? a,b,c is not explained, but is showing the activation of hydrophobic amino acid through the phosphopanthetein-transferase, whichis more general, followed by the condensation through peptide bond formation, but this is the general understanding, right (i.e. Marahiel et al. 1997)
L380: Suomilide has been described in 1997 (not in 2021)
References
Suda S, Watanabe MM, Otsuka S, Mahakahant A, Yongmanitchai W, Nopartnaraporn N, Liu Y, Day JG. Taxonomic revision of water-bloom-forming species of oscillatorioid cyanobacteria. Int J Syst Evol Microbiol. 2002 Sep;52(Pt 5):1577-1595. doi: 10.1099/00207713-52-5-1577. PMID: 12361260.
(2007) Biosynthesis and structure of aeruginoside 126A and 126B, cyanobacterial peptide glycosides bearing a 2-carboxy-6-hydroxyoctahydroindole moiety. Chem Biol 14(5), 565-576.
Heinilä, Lassi & Jokela, Jouni & Ahmed, Muhammad & Wahlsten, Matti & Saurav, Kumar & Hrouzek, Pavel & Permi, Perttu & Koistinen, Hannu & Fewer, David & Sivonen, Kaarina. (2022). Discovery of varlaxins, new aeruginosin-type inhibitors of human trypsins. Organic & Biomolecular Chemistry. 20. 10.1039/D1OB02454J.
Fujii, Kiyonaga & Sivonen, Kaarina & Naganawa, Emiko & Harada, Ken-ichi. (2000). Non-Toxic Peptides from Toxic Cyanobacteria, Oscillatoria agardhii. Tetrahedron. 56. 725-733. 10.1016/S0040-4020(99)01017-0.
Marahiel MA, Stachelhaus T, Mootz HD. Modular Peptide Synthetases Involved in Nonribosomal Peptide Synthesis. Chem Rev. 1997 Nov 10;97(7):2651-2674. doi: 10.1021/cr960029e. PMID: 11851476.
Reviewer 2 Report
The article is well written. Authors present some aspects of aeruginosins, including structural characterization, biosynthesis and bioactivity. First are described 29 compounds, and are shown chemical structures. Next is presented structure diversity with impact of residues and modifications of Choi moiety. In Bioactivity are described inhibitory effects to trypsin, thrombin and plasmin, and anti-inflammatory activity. The authors exhausted the topic by citing 85 references. In PubMed, after typing in the entry "aeruginosins", there are only 117 articles.
Only correction, which I suggest is use "aeruginosins" in the title, instead of "aeruginosin" (Diversity, Biosynthesis and Bioactivity of Aeruginosins...).
Reviewer 3 Report
This is very interesting review paper, well-presented and preared. I have only few issues to be amended.
1. Unit of IC50 values should be unified to uM to compare their inhibitory activity.
2. Side effect or toxicity issues should be added.
3. In vivo profiles (efficacy and pharmacokinetics, etc.) of these compounds should be added.
4. Limits or merits of these compounds to be developed as drug should be also mentioned.
Reviewer 4 Report
The manuscript is interesting. This is a review work. Is a comprehensive review of aeruginsins. In this review, providing an overview of the source, chemical structure as well as the spectrum of bioactivities of aeruginsins. The manuscript makes a significant contribution to the area of the discipline it represents. The issues are exhaustively described. There are also relevant literature references to earlier works.
It also provides guidelines for future research.
The publication should be published after making some minor corrections.
1. Keywords should not coincide with the phrases listed in the manuscript title.
2. Please write Latin names in italics, e.g. line 42,43.
3. Please correct the citation of literature - now the order is not preserved. Please see, for example, lines 42-64.
4. Please, be sure that all the references are included in the reference list and vice versa with matching spellings and dates.
Round 2
Reviewer 3 Report
Authors have fully addressed all issues. Therefore, this paper is now acceptable.